# Shaofu Zhuyu Decoction for Treating Endometriosis: A Systematic Review and Meta-Analysis

**DOI:** 10.3390/ph18091296

**Published:** 2025-08-29

**Authors:** Su-Bin Kim, Young-Jin Yoon, Soo-Hyun Sung, Jang-Kyung Park

**Affiliations:** 1Department of Korean Medicine, School of Korean Medicine, Pusan National University, Yangsan 50612, Republic of Korea; jenny011122@naver.com; 2Department of Korean Medicine Obstetrics and Gynecology, School of Korean Medicine, Pusan National University, Yangsan 50612, Republic of Korea; yyj@pusan.ac.kr; 3Department of Obstetrics and Gynecology of Korean Medicine, Pusan National University Korean Medicine Hospital, Yangsan 50612, Republic of Korea; 4Department of Policy Development, National Development Institute for Korean Medicine, Seoul 04516, Republic of Korea

**Keywords:** Shaofu Zhuyu decoction, endometriosis, Chinese herbal medicine, meta-analysis, traditional medicine, CA-125, pain, recurrence, systematic review

## Abstract

**Introduction**: Shaofu Zhuyu Decoction (SZD) is a traditional Chinese herbal formula commonly used for gynecological disorders involving “blood stasis” and lower abdominal pain. Although applied clinically for endometriosis, evidence regarding its efficacy and safety remains fragmented. This systematic review and meta-analysis aimed to evaluate the clinical efficacy and safety of SZD combined with conventional medication (CM) for the treatment of endometriosis. **Methods**: Databases including PubMed, EMBASE, CNKI, and Web of Science were searched from inception to May 2024 for randomized controlled trials (RCTs) comparing SZD + CM versus CM alone. Risk of bias was assessed using the risk of bias 2.0 tool. The primary outcome was total effective rate (TER); secondary outcomes included serum CA-125 levels, pain scores (VAS), recurrence rate, and adverse events. **Results**: Eleven RCTs (*n* = 1186 patients) were included. Meta-analysis showed that SZD plus CM significantly improved TER compared to CM alone (OR 1.15; 95% CI: 1.09–1.22; *p* < 0.00001). Reductions in CA-125 levels (OR −1.57; 95% CI: −2.18 to −0.95; *p* < 0.00001) and pain (VAS) scores (OR −4.90; 95% CI: −6.82 to −2.98; *p* < 0.00001) were also significant. Three studies reported lower recurrence in the SZD group. Adverse events were generally mild and comparable between groups. **Conclusions**: SZD combined with CM appears more effective than CM alone in improving endometriosis symptoms, reducing biochemical markers, and decreasing pain intensity, with acceptable safety. However, the findings are limited by methodological heterogeneity and geographic concentration of studies. Rigorous multicenter trials are warranted to support integration of SZD into global endometriosis care.

## 1. Introduction

Endometriosis is a chronic inflammatory disease defined by the presence of endometrial-like tissue outside the uterine cavity, affecting approximately 10–15% of women of reproductive age and commonly presenting with pelvic pain, dysmenorrhea, and infertility [1,2,3]. While the precise pathogenesis remains unclear, multiple theories—such as retrograde menstruation, immune dysfunction, and stem cell involvement—have been proposed to explain its development [3,4]. Diagnosis typically relies on laparoscopic examination due to the heterogeneous and often nonspecific nature of symptoms, resulting in a prolonged diagnostic delay [5].

Current treatment strategies for endometriosis vary depending on the patient’s age, reproductive plans, and symptom severity. Pharmacological therapy is generally considered the first-line approach, while surgical intervention is reserved for cases with severe or refractory symptoms. Hormonal agents such as progestogens, danazol, and gonadotropin-releasing hormone (GnRH) agonists and antagonists are commonly used in medical treatment. These therapies can provide temporary symptom relief; however, they are often associated with adverse effects, including weight gain, increased risk of osteoporosis, acne, and voice changes, and they rarely achieve complete disease resolution [3]. Surgical treatment options include excision of localized lesions or resection of the uterus and/or ovaries. However, recurrence rates can be as high as 50% if postoperative hormonal therapy is not administered [6,7,8].

Given the limitations of existing therapeutic options, complementary and alternative medicine (CAM), particularly traditional Chinese medicine (TCM), has attracted increasing global attention. TCM has long been applied in East Asia for gynecological disorders, and emerging evidence indicates its potential to alleviate clinical symptoms while addressing underlying pathophysiological mechanisms. Among the various TCM formulas, Shaofu Zhuyu Decoction (SZD) is commonly prescribed for endometriosis due to its traditional functions of promoting blood circulation, removing blood stasis, and relieving pain.

SZD is a traditional Chinese herbal formula originating from the Yilin Gaicuo (Corrections of Errors in Medical Classics, Qing dynasty, 1830). It consists of ten medicinal herbs, including *Foeniculi Fructus*, *Zingiberis Rhizoma*, *Corydalis Rhizoma*, and *Ligustici Chuanxiong Rhizoma* [9]. The formula is traditionally used in gynecology to relieve lower abdominal pain caused by blood stasis, based on its pharmacological actions of promoting blood circulation, resolving stasis, warming the meridians, and alleviating pain [10]. Clinically, SZD has been applied to the management of primary dysmenorrhea [11], chronic pelvic inflammatory disease [12], and uterine fibroids [13]. Its analgesic effects have been attributed to its anti-inflammatory properties and its ability to inhibit uterine smooth muscle contraction [14,15]. Preclinical studies exploring the therapeutic mechanisms of SZD in endometriosis have demonstrated its anti-inflammatory and analgesic potential [10,16].

To date, several randomized controlled trials (RCTs) have evaluated the use of SZD in patients with endometriosis. However, the interpretability of these findings is limited due to small sample sizes and heterogeneity in outcome measures. Moreover, existing meta-analyses have primarily focused on combination therapy involving SZD and conventional medicine (CM), making it difficult to isolate the effects of SZD monotherapy [17,18]. Notably, no comprehensive systematic review and meta-analysis has yet been conducted to evaluate the overall efficacy and safety of SZD in patients with endometriosis. Therefore, the present study aims to systematically analyze RCTs, evaluating both SZD monotherapy and combination therapy. The goal is to comprehensively assess clinical outcomes in endometriosis patients, including symptom improvement, reproductive health indicators, inflammatory and hormonal biomarkers, and adverse events.

## 2. Materials and Methods

### 2.1. Design

This systematic review and meta-analysis aimed to assess the efficacy and safety of SZD in the treatment of endometriosis. The review process adhered to the guidelines outlined in the Preferred Reporting Items for Systematic Reviews and Meta-Analyses (PRISMA) statement [19], with the completed PRISMA checklist provided in Appendix A. The review protocol was prospectively registered in the International Prospective Register of Systematic Reviews (PROSPERO) under the registration number CRD420251102092 (https://www.crd.york.ac.uk/PROSPERO/view/CRD420251102092, accessed on 1 August 2025).

### 2.2. Data Sources and Searches

A comprehensive literature search was performed across 11 electronic databases through March 2025. The databases included MEDLINE, EMBASE, the Cochrane Central Register of Controlled Trials (CENTRAL), China National Knowledge Infrastructure (CNKI), Wanfang Data, ScienceON, the Korean Traditional Knowledge Portal, KoreaMed, the Oriental Medicine Advanced Searching Integrated System (OASIS), the Research Information Sharing Service (RISS), and the National Library of Korea.

The search strategy combined terms related to the intervention and condition of interest: ((“Shaofu Zhuyu decoction” OR “Shaofu Zhuyu Tang” OR “Shaofu Zhuyu formula”) AND “Endometriosis” AND (“clinical trial” OR “randomized controlled trial” OR “randomised controlled trial”)). Detailed search strategies tailored to each database are provided in Appendix A.

### 2.3. Inclusion and Exclusion

#### 2.3.1. Participants

Studies enrolling patients diagnosed with endometriosis, without restriction on age, ethnicity, or other demographic factors, were included.

#### 2.3.2. Types of Interventions

Studies were included if SZD was administered either as a monotherapy or in combination with CM. To accurately assess the effect of SZD, studies involving combination therapy (SZD plus CM) were included only when the control group received the same CM alone. There were no restrictions on the composition or dosage of SZD; studies were included if SZD was prescribed to patients. This approach was adopted because the composition and dosage of the herbal formula may be modified according to patient symptoms or syndrome differentiation, and these details were summarized separately.

#### 2.3.3. Types of Comparisons

RCTs comparing SZD with all types of control interventions were included in the review.

#### 2.3.4. Types of Outcomes

Primary outcomes considered (1) total effective rate (TER), (2) tumor maker levels (e.g., CA-125), and (3) pain index (e.g., VAS). Secondary outcomes included (1) recurrence rate, (2) AE occurrence rate, (3) inflammatory factors (e.g., TNF-α, PGF2α), (4) quality of life, (5) neuroangiogenic factors (e.g., bFGF, VEGF), and (6) adverse events.

#### 2.3.5. Types of Studies

The eligibility criteria for study selection were as follows: (1) availability of the full text, (2) publication in either English or Korean, (3) original articles published in peer-reviewed journals, and (4) RCT design. Studies were excluded if they were (1) abstracts from conferences, letters, or commentaries; (2) published in languages other than English; (3) clinical studies that were not RCTs, such as case reports, case series, or case-control studies; (4) animal or laboratory-based (in vitro) research; or (5) reviews, qualitative research, survey studies, or study protocols.

### 2.4. Study Selection and Data Extraction

Two independent reviewers (S.-B.K. and Y.-J.Y.) screened the titles and abstracts of all identified records according to predefined inclusion and exclusion criteria. The included studies were then characterized using the PICO framework (P: patient, I: intervention, C: comparator, O: outcome). Data extraction was independently performed by two reviewers (S.-B.K. and S.-H.S.) utilizing a standardized data collection form. Furthermore, the same reviewers independently collected detailed information on authorship, sample sizes, interventions, outcome measures, primary results, and adverse events. Specific details regarding the composition of the SZD intervention were also documented. Throughout the process, all data were cross-checked among the reviewers, and discrepancies were resolved by discussion. When consensus could not be reached, the corresponding author (J.-K.P.) was consulted to make the final decision.

### 2.5. Assessment of Risk of Bias (ROB)

The risk of bias for included studies was independently evaluated by two reviewers (S.-B.K. and J.-H.S.) using the Cochrane Collaboration Risk of Bias (ROB) 2.0 tool [20]. This assessment encompassed six domains: (1) the randomization process, (2) deviations from intended interventions, (3) missing outcome data, (4) outcome measurement, (5) selection of reported results, and (6) the overall risk of bias. Each domain was categorized as low risk, high risk, or presenting some concerns. Any disagreements between reviewers were discussed and resolved by consensus.

### 2.6. Data Analyses

Data extracted from the included studies were synthesized through qualitative methods and, when appropriate, quantitative meta-analysis. For studies reporting comparable outcomes and utilizing similar methodologies, meta-analyses were conducted using Review Manager (RevMan) version 5.4 (Nordic Cochrane Center, Copenhagen, Denmark). Effect estimates were expressed as risk ratios (RRs) or odds ratios (ORs) for dichotomous variables, and as mean differences (MDs) or standardized mean differences (SMDs) for continuous variables, all accompanied by 95% confidence intervals (CIs). To address potential heterogeneity among studies, a random effects model was employed.

Statistical heterogeneity was evaluated using the I^2^ statistic, with values of 25%, 50%, and 75% interpreted as low, moderate, and high heterogeneity, respectively. In cases where meta-analysis was deemed inappropriate due to substantial clinical or methodological heterogeneity, a narrative synthesis was performed, organized by intervention type, population characteristics, and measured outcomes.

## 3. Results

### 3.1. Study Selection and Description

A total of 110 records were initially identified. After screening titles and abstracts, 8 studies unrelated to SZD or endometriosis and 13 experimental (non-clinical) studies were excluded. Full texts of the remaining 89 articles were assessed, resulting in the exclusion of 59 non-randomized studies and 19 studies not involving SZD as an intervention. Ultimately, 11 RCTs [21,22,23,24,25,26,27,28,29,30,31] were included in this review. The study selection process is illustrated in Figure 1. All included studies were conducted in China and published in Chinese. The detailed characteristics of the included studies are summarized in Table 1.

### 3.2. Participants

A total of 1186 patients with endometriosis were included across 11 RCTs, with 593 patients assigned to the experimental group and 593 to the control group.

### 3.3. Intervention

Among the 11 included RCTs, 10 studies [21,22,23,24,25,26,27,28,29,30] compared SZD plus conventional treatment with conventional treatment alone, while 1 study [31] compared SZD monotherapy with conventional treatment. Details regarding the traditional Chinese medicine (TCM) pattern identification, herbal composition of SZD, modified ingredients, dosage, and duration of administration in the experimental groups are summarized in Table 2. In the control groups, seven types of CMs were used, including progesterone (three studies) [22,28,30], mifepristone (two studies) [24,31], ibuprofen (two studies) [25,27], danazol [21], dinoprostone [29], gestrinone [23], and either GnRH-a or Mirena ring [26], each used in one study. The specific names, dosages, and durations of the CMs are detailed in Table 3.

#### 3.3.1. Herbs Composing Shaofu Zhuyu Decoction in the Included Studies

Among the included studies, seven trials [21,22,23,24,25,26,27] used SZD composed of the original ten herbs, while three studies [28,29,30] used fewer than ten herbs, and one study [31] employed a modified formula with more than ten herbs. An analysis of individual herb usage revealed that *Angelicae Gigantis Radix*, *Cinnamomi Cortex*, *Cnidii Rhizoma*, *Corydalis Tuber*, *Foeniculi Fructus*, *Trogopterori Faeces*, *Typhae Pollen*, and *Zingiberis Rhizoma* were included in all studies. *Paeoniae Radix Rubra* was used in ten studies [21,22,23,24,25,26,27,28,30,31], while *Myrrha* appeared in eight studies [21,22,23,24,25,26,27,31]. In addition to the core SZD formula, several studies included supplementary herbs; *Curcumae Rhizoma*, *Eupolyphaga*, *Fluoritum*, and *Sparganii Rhizoma* were each used in one study [31] (Table 2).

#### 3.3.2. Modified Herbs

In eight studies, the composition of SZD was modified based on accompanying symptoms or the severity of the condition. For patients experiencing abdominal pain, *Achyranthis Radix*, *Rhododendri Mollis Cortex*, and *Dragon’s Blood* were added [28,29,30]. When abdominal pain was accompanied by bloating, *Angelicae Dahuricae Radix* and *Toosendan Fructus* were added [23], and in cases of cold-type abdominal pain, *Citri Reticulatae Semen* and *Litchi Semen* were used [23].

In the presence of pelvic nodules or masses, *Sparganii Rhizoma* and *Curcumae Rhizoma* [21,22,24,26], *Gleditsiae Sinensis Fructus* and *Persicae Semen* [22,24,26], *Aucklandiae Radix* [23], *Carthami Flos* [22], and *Eupolyphaga* [21] were added. For cases involving ectopic cysts on the endometrium, *Persicae Semen*, *Vaccariae Semen*, *Hirudo*, and *Rhei Radix et Rhizoma* were added [23]. In patients with menorrhagia, *Rubiae Radix et Rhizoma* and *Notoginseng Radix* [23,24,26,28,30], *Sanguisorbae Radix* and *Os Draconis* [24,26], and *Cephalanoplos Herba* and *Cacaliae Herba* [29] were used. When menstrual volume was scant, *Carthami Flos*, *Persicae Semen*, and *Chuanxiong Rhizoma* were added [23]. For prolonged uterine bleeding, *Artemisiae Argyi Folium*, *Zingiberis Rhizoma Carbonisatum*, and *Leonuri Herba* were used [21,22]. In anemic patients, *Astragali Radix* and *Angelicae Gigantis Radix* were added [29], while *Pinelliae Tuber* and *Citri Unshius Pericarpium* were added in cases with nausea or vomiting [23].

In cases with severe symptoms, herbal modifications were made based on syndrome differentiation. For pronounced cold dampness, *Zingiberis Rhizoma*, *Zingiberis Rhizoma Carbonisatum*, and *Evodiae Fructus* were added [28,30]. For marked liver qi stagnation, *Bupleuri Radix* and *Curcumae Radix* were prescribed [28,30]. In cases of severe spleen deficiency, *Atractylodis Macrocephalae Rhizoma* and *Atractylodis Rhizoma* were used [28,30], while for pronounced spleen–kidney yang deficiency, *Eucommiae Cortex* and *Codonopsis Radix* were added [28,30].

For patients with significant qi deficiency, *Polygonati Rhizoma* and *Astragali Radix* were included [29]. When blood stasis was dominant, *Sparganii Rhizoma* and *Curcumae Rhizoma* were used [28,30]. In cases of severe yang deficiency, *Psoraleae Semen*, *Aconiti Lateralis Radix Praeparata*, and *Morindae Officinalis Radix* were added [21]. For infertility associated with congenital deficiency of essence and weakness of the Chong and Ren meridians, *Cuscutae Semen*, *Cnidii Fructus*, *Cynomorii Herba*, and *Trachelospermi Caulis* were prescribed [21] (Table 2).

### 3.4. Outcomes

#### 3.4.1. SZD Plus CM Versus CM

A total of 10 studies compared SZD combined with CM versus CM alone. For the primary outcome, seven clinical trials [21,22,23,24,25,26,29] assessed TER. Meta-analysis showed that the SZD plus CM group demonstrated a statistically significant improvement compared to the CM group (OR = 1.15; 95% CI: 1.09–1.22; *p* < 0.00001) (Figure 2a). In five studies [23,24,28,29,30], meta-analysis revealed a significant reduction in serum CA-125 levels in the experimental group compared to the control group (OR = −1.57; 95% CI: −2.18 to −0.95; *p* < 0.00001) (Figure 2b). Additionally, four studies [25,27,28,30] reported that the combination of SZD and CM was significantly more effective than CM alone in alleviating pain (OR = −4.90; 95% CI: −6.82 to −2.98; *p* < 0.00001) (Figure 2c).

For the secondary outcomes, all four trials [21,22,24,26] reported that the recurrence rate was significantly lower in the SZD plus CM group compared to the CM group alone (*p* < 0.05). Inflammatory markers such as tumor necrosis factor-alpha (TNF-α) and prostaglandin F2 alpha (PGF2α) were significantly reduced in all three studies that assessed them [24,25,27]. In the meta-analysis of two trials [28,30], the combination of SZD and CM significantly improved quality of life compared to CM alone (OR = −1.12; 95% CI: −1.44 to −0.79; *p* < 0.00001) (Figure 3a). Furthermore, a meta-analysis of two studies reported statistically significant improvements in both neuroangiogenic factors—basic fibroblast growth factor (bFGF) and vascular endothelial growth factor (VEGF) (Figure 3b,c).

#### 3.4.2. SZD Versus CM

One RCT [31] compared SZD to CM showed statistically significant results from pain index, TER, tumor marker, inflammatory factor, and immunological factor (*p* < 0.05).

### 3.5. Adverse Events

AEs were reported in five studies [22,23,24,25,31]. One study [24] reported no AEs in either group. Across the remaining four studies, 17 participants in the experimental group and 14 in the control group experienced adverse events. In the experimental group, the reported AEs were mostly mild and included cold sweats, skin pruritus, mild skin rash, and skin pruritus. One case of vaginal bleeding was reported [22], but it did not require medical intervention. Similarly, the AEs reported in the control group were predominantly mild, including facial flushing, skin pruritus, and night sweats. Some participants in two studies [22,31] reported irregular vaginal bleeding and palpitations; however, these symptoms were also classified as mild, as no therapeutic measures were required. Detailed adverse events from the five studies are summarized in Table 1.

### 3.6. Assessment for ROB

The overall ROB for the included studies [21,22,23,24,25,26,27,28,29,30,31] was assessed as follows: ten studies (90.9%) were rated as having “some concerns”, and one study (9.1%) was judged to have a high risk of bias (Figure 4). Ten studies [21,22,23,24,25,26,27,29,30,31] were rated as having “some concerns” due to insufficient information for assessing the domains of the randomization process, deviations from intended interventions, measurement of the outcome, and selection of the reported result. In the domain of missing outcome data, all ten studies were considered to have a low risk of bias, as they either reported no dropout or reported dropout rates of approximately 5%. The one remaining study used an odd–even allocation method, which cannot be regarded as an appropriate randomization method, and did not provide information regarding allocation concealment. Therefore, this study was rated as having a high risk of bias in the randomization process and an overall risk of bias.

## 4. Discussion

### 4.1. Main Finding and Its Implication

This systematic review and meta-analysis assessed the clinical efficacy and safety of SZD combined with CM for managing endometriosis, drawing on data from 11 RCTs. The results consistently demonstrated that the combination of SZD and CM yielded better clinical outcomes than CM alone.

The TER, which serves as a primary indicator of overall symptom improvement, was significantly higher in the SZD plus CM groups across seven trials [21,22,23,24,25,26,29]. The pooled meta-analysis revealed a statistically significant advantage in TER for the combination therapy (*p* < 0.00001), suggesting that integrating traditional herbal formulas may enhance therapeutic effects beyond those achieved with conventional treatments alone. This finding aligns with prior meta-analyses highlighting the potential benefits of Chinese herbal medicine in endometriosis management [32]. Regarding pain reduction, measured by the VAS and other dysmenorrhea severity assessments, four trials [25,27,28,30] reported significant improvements in the SZD-combined groups. Meta-analytic synthesis confirmed a marked decrease in pain scores (*p* < 0.00001). Given the substantial impact of pain on quality of life in endometriosis patients, these findings are clinically important and may reflect the multifaceted mechanisms of SZD, including its anti-inflammatory, analgesic, and blood circulation-promoting properties. These findings are in line with previous studies demonstrating that combining herbal interventions with CMs results in enhanced pain relief [32].

Serum CA-125, a well-established biomarker reflecting endometriosis severity, showed a significant decrease in the intervention group. Specifically, five randomized controlled trials [23,24,28,29,30] reported greater reductions in CA-125 levels when SZD was combined with CM compared to CM alone (*p* < 0.00001), indicating the potential biochemical impact of SZD on disease-associated inflammatory or tumor markers. This meta-analysis indicates that SZD, when combined with CMs, may alleviate clinical symptoms of endometriosis, including pain and elevated serum CA-125 levels. However, the proposed mechanisms of action remain largely hypothetical and are not directly supported by the current clinical evidence. Accordingly, the potential disease-modifying effects of SZD should be interpreted with caution. Adverse events were infrequent and comparable between groups; they were predominantly mild (including skin reactions, flushing, and gastrointestinal symptoms), with no serious adverse events reported, supporting a favorable safety profile consistent with previous clinical reports.

### 4.2. Strength and Limitation of Study

This review represents the first systematic effort to comprehensively evaluate both the efficacy and safety of SZD in the management of endometriosis. In contrast to earlier meta-analyses that largely focused on SZD combined with CM, our study also considers SZD monotherapy and a broader spectrum of clinical endpoints. By adopting this approach, the present work offers a more integrated perspective on the therapeutic potential of SZD. While this paper does not report original experimental data, its novelty lies in synthesizing evidence from multiple randomized controlled trials using standardized systematic review methodology, thereby generating new insights into efficacy and safety across diverse treatment regimens.

This study has several limitations. One of the most significant limitations of this meta-analysis is the high degree of heterogeneity in the control interventions across the included studies [33]. Specifically, the control groups encompassed seven distinct CMs—progesterone, mifepristone, ibuprofen, danazol, dinoprostone, gestrinone, and GnRH-a/Mirena—each with different mechanisms of action, efficacy profiles, and clinical indications. Pooling studies with such heterogeneous comparators into a single meta-analysis, without conducting subgroup analyses according to medication type, may substantially obscure nuanced differences in treatment effects and limits the interpretability of the combined results. Addressing this limitation will require a greater number of rigorously designed RCTs, allowing subgroup analyses stratified by the type of control intervention. Second, despite including international databases in the search strategy, all 11 included RCTs were conducted in China and published in Chinese. This reflects the historical and practical context of SZD as a traditional Chinese herbal medicine, rather than a lack of research rigor. Nevertheless, it may restrict the generalizability of our conclusions to broader international populations and could introduce regional research or reporting bias [34,35]. Third, the overall methodological quality of the included trials was rated as moderate to low based on the Cochrane Risk of Bias tool. Therefore, caution is necessary when interpreting the results, as methodological rigor alone does not fully address issues related to external validity or potential subjectivity in outcome assessments. Finally, only four of the included studies reported adverse events, highlighting the need for more comprehensive safety data. Monitoring and reporting adverse events are crucial for establishing treatment safety and guiding clinical recommendations [36]. Future RCTs investigating SZD should prioritize thorough adverse event reporting and develop detailed safety profiles to facilitate its informed clinical use.

### 4.3. Future Perspective

All the studies included in this meta-analysis were conducted in China, often with relatively small sample sizes and potential regional treatment biases. To confirm the global applicability of SZD in terms of efficacy and safety, large-scale, multinational, and multicenter randomized controlled trials involving diverse ethnicities, genetic backgrounds, and clinical characteristics are urgently needed. Furthermore, standardization of SZD prescription according to symptom patterns in patients with dysmenorrhea is essential. This process is critical for accurately evaluating SZD’s therapeutic effects and safety, as well as for developing clinical practice guidelines and facilitating pharmaceutical formulation. Following prescription standardization, guidelines for combined SZD and CM use should be established.

Although clinical improvements—such as symptom relief and reductions in CA-125—are well documented, the molecular and cellular mechanisms underpinning SZD’s efficacy in endometriosis remain poorly understood. Employing systems biology approaches, metabolomics, and gut microbiota profiling could uncover novel therapeutic targets and clarify SZD’s modulatory effects on inflammation, immune responses, and neuroangiogenesis. Future investigations should also prioritize patient-reported outcomes, including quality of life, mental health status, and functional capacity, as primary endpoints. Additionally, qualitative research addressing patient preferences, cultural factors, and barriers to integrative treatment will provide valuable insights into the feasibility of SZD’s broader implementation. Exploration of shared decision-making frameworks and multidisciplinary care models is warranted. Given that dysmenorrhea is a recurrent monthly challenge significantly impacting women’s lives, research should extend beyond quantitative measures to include comprehensive assessments of quality of life, mental well-being, interpersonal relationships, and effects on academic or occupational functioning. Such holistic evaluation can deepen clinicians’ understanding and improve patient counseling, while providing policy-makers with evidence to support targeted interventions.

In summary, advancing the clinical application of SZD in endometriosis will require future studies emphasizing methodological rigor, mechanistic elucidation, and patient-centered outcomes to ensure its safe, effective, and culturally sensitive integration into global management strategies.

## 5. Conclusions

These results support the integrative use of SZD alongside hormonal or pharmacological CM regimens in managing endometriosis. This combined approach may provide a multi-targeted therapeutic effect by harnessing the anti-inflammatory, analgesic, and blood stasis resolving properties of traditional herbal medicine, thereby complementing the endocrine modulation offered by CM drugs.

However, despite these encouraging outcomes, the current evidence is constrained by methodological limitations, variability in SZD formulations, and relatively short follow-up periods, with all included trials conducted exclusively in China. To strengthen the evidence base, larger, rigorously designed, and multicenter studies involving diverse populations are necessary. Should such studies confirm its efficacy and safety, SZD could become a valuable adjunct within integrative treatment frameworks and would be particularly appealing to patients seeking holistic and sustained symptom management strategies.

## Figures and Tables

**Figure 1 pharmaceuticals-18-01296-f001:**
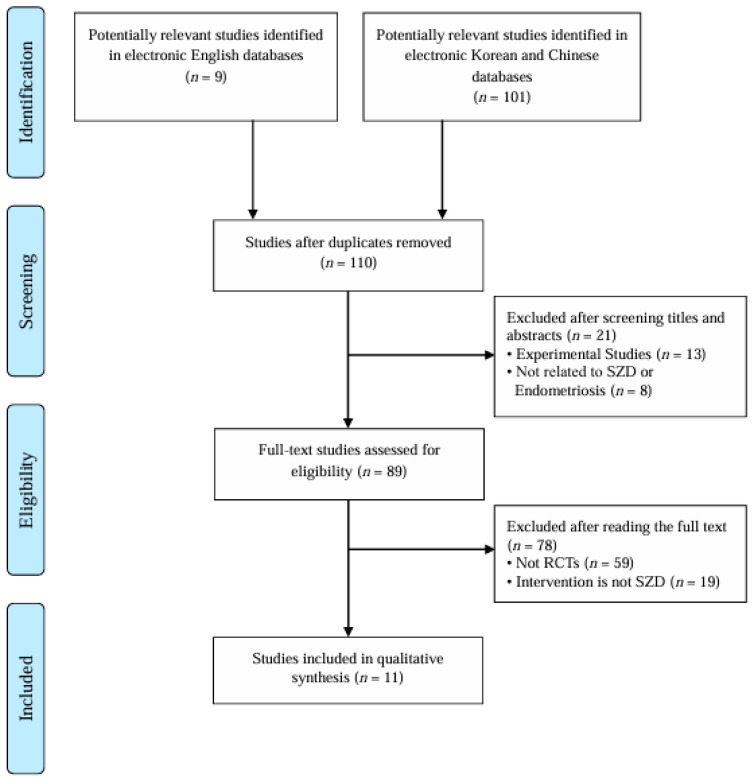
PRISMA study flow diagram. RCTs: randomized controlled trials, SZD: Shaofu Zhuyu Decoction.

**Figure 2 pharmaceuticals-18-01296-f002:**
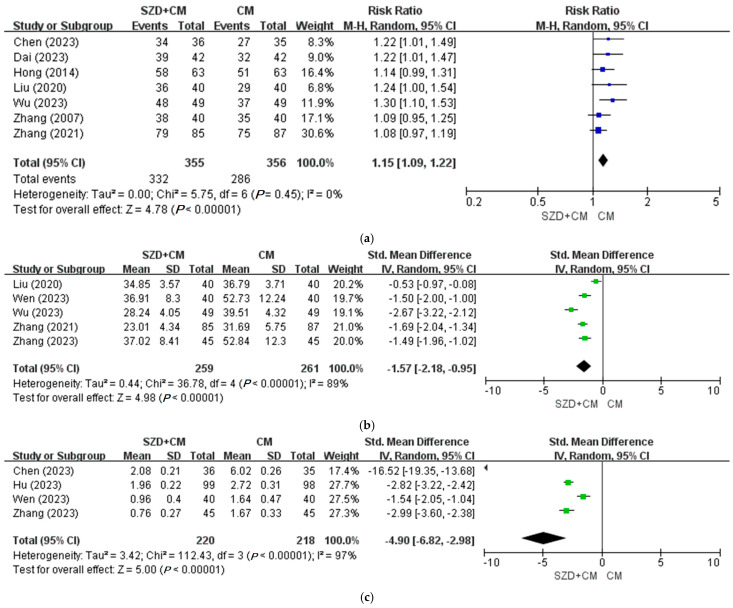
Meta-analysis of Shaofu Zhuyu Decoction for endometriosis in primary outcomes. (**a**) SZD plus CM versus CM, TER [21,22,23,24,25,26,29]; (**b**) SZD plus CM versus CM, CA-125 [23,24,28,29,30]; (**c**) SZD plus CM versus CM, VAS [25,27,28,30]. CA-125: Cancer Antigen 125, CI: Confidence Interval, CM: Conventional Medication, SZD: Shaofu Zhuyu Decoction, TER: Total Effective Rate, VAS: Visual Analog Scale.

**Figure 3 pharmaceuticals-18-01296-f003:**
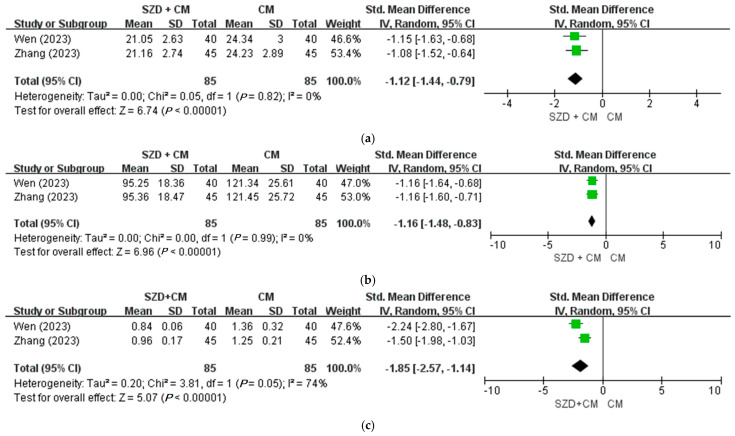
Meta-analysis of Shaofu Zhuyu Decoction for endometriosis in secondary outcomes. (**a**) SZD plus CM versus CM, quality of life (EHP) [28,30]; (**b**) SZD plus CM versus CM, neuroangiogenic factor (bFGF) [28,30]; (**c**) SZD plus CM versus CM, neuroangiogenic factor (VEGF) [28,30]. CI: Confidence Interval, CM: Conventional Medication, EHP: Endometriosis Health Profile, SZD: Shaofu Zhuyu Decoction.

**Figure 4 pharmaceuticals-18-01296-f004:**
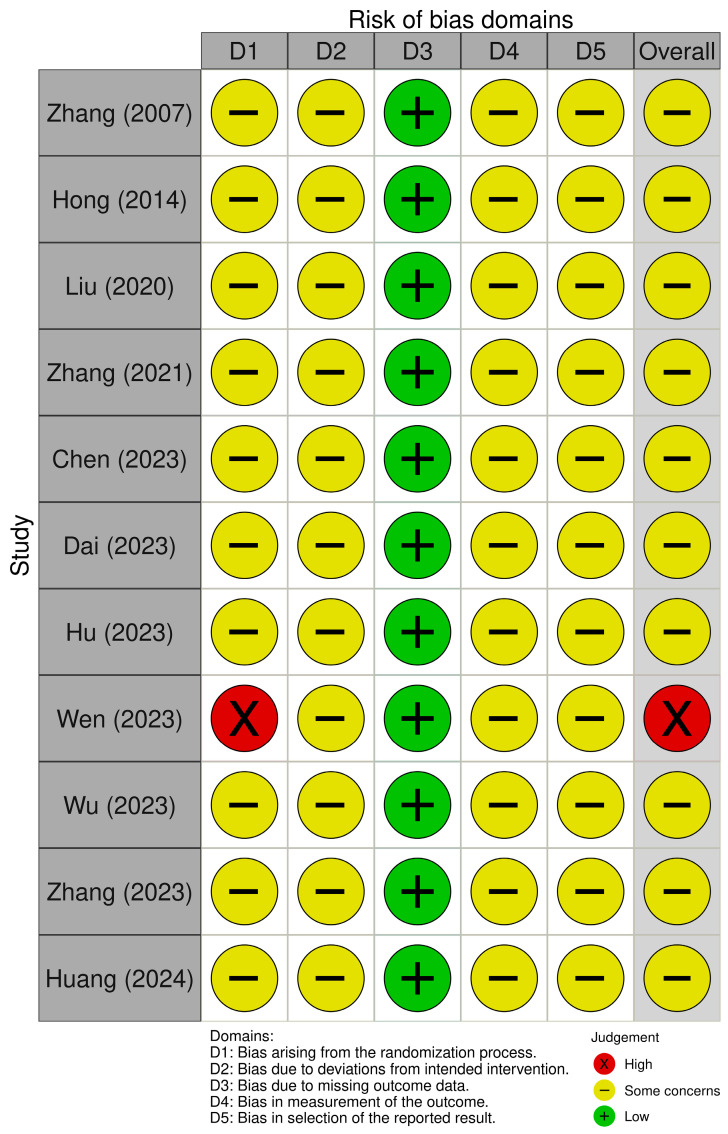
Risk of bias assessment [21,22,23,24,25,26,27,28,29,30,31].

**Table 1 pharmaceuticals-18-01296-t001:** Characteristics of the included studies.

Study ID (Author, Year)	Experimental Group (No. of Participants Analyzed/Randomized)	Experimental Group	Duration	F/U	Control Group (No. of Participants Analyzed/Randomized)	Control Group	Duration	F/U	Outcome Measurement	Results	Adverse Event
Zhang (2007) [21]	40/40	Shaofu Zhuyu Decoction + Danazol	6 months + 3 months (if not treated)	1 year	40/40	Danazol	6 months	1 year	1. TER 2. Recurrence rate	1. positive (*p* < 0.05) 2. positive (*p* < 0.05)	N.R.
Hong (2014) [22]	63/63	Shaofu Zhuyu Decoction + Progesterone	3 months	1~2 years	63/63	Progesterone	3 months	1~2 years	1. TER 2. Recurrence rate 3. AE occurrence rate	1. positive (*p* < 0.05) 2. positive (*p* < 0.01) 3. NS	(1) SZD + CM group: Cold sweats (2 cases, 3.17%), Skin pruritus (2 cases, 3.17%), Facial flushing (2 cases, 3.17%), Vaginal bleeding (1 case, 6.35%), Vaginal dryness (1 case, 6.35%), Amenorrhea (1 case, 6.35%) (2) CM group: Cold sweats (2 cases, 6.35%), Facial flushing (2 cases, 6.35%), Irregular vaginal bleeding (2 cases, 3.17%), Vaginal dryness (1 case, 1.59%), Mild gastrointestinal reaction (1 case, 1.59%)
Liu (2020) [23]	40/40	Shaofu Zhuyu Decoction + Gestrinone	3 months	None	40/40	Gestrinone	3 months	None	1. TER 2. Tumor maker level (1) CA-125 3. AE occurrence rate	1. positive (*p* < 0.05) 2. (1) positive (*p* < 0.05) 3. NS	(1) SZD + CM group: Mild skin rash (2 cases) (2) CM group: Mild nausea (1 case)
Zhang (2021) [24]	85/90	Shaofu Zhuyu Decoction + Mifepristone	3 months	1 year	87/90	Mifepristone	3 months	1 year	1. TER 2. Tumor maker level (1) CA125 3. Inflammatory factor levels (1) PGE2 (2) PGF 2α 4. Recurrence rate	1. No significant difference 2. (1) positive (*p* < 0.05) 3. (1) positive (*p* < 0.05) (2) positive (*p* < 0.05) 4. positive (*p* < 0.05)	No AEs occurred in (E) and (C)
Chen (2023) [25]	36/36	Shaofu Zhuyu Decoction + Ibuprofen	3 months	None	35/35	Ibuprofen	3 months	None	1. Pain index (1) VAS 2. TER 3. Inflammatory factor level (1) PGF2α	1. (1) positive (*p* < 0.05) 2. positive (*p* < 0.05) 3. (1) positive (*p* < 0.05)	(1) SZD + CM group: Facial flushing (2 cases), Skin pruritus (1 case), Irregular vaginal bleeding (1 case) (2) CM group: Facial flushing (1 case), Skin pruritus (1 case), Night sweats (1 case)
Dai (2023) [26]	42/42	Shaofu Zhuyu Decoction + GnRH-a or Mirena ring	3 months	1 year	42/42	GnRH-a or Mirena ring	3 months	1 year	1. TER 2. Recurrence rate	1. positive (*p* < 0.05) 2. positive (*p* < 0.05)	N.R.
Hu (2023) [27]	99/99	Shaofu Zhuyu Decoction + Ibuprofen	3 months	None	98/98	Ibuprofen	3 months	None	1. Pain index (1) VAS (2) Pain duration (3) Dysmenorrhea symptom score 2. Inflammatory factor levels (1) TNF-α	1. (1) positive (*p* < 0.05) (2) positive (*p* < 0.05) (3) positive (*p* < 0.05) 2. (1) positive (*p* < 0.05)	N.R.
Wen (2023) [28]	40/40	Shaofu Zhuyu Decoction + Progesterone	3 months	1 year	40/40	Progesterone	3 months	1 year	1. Pain index (1) VAS 2. Tumor maker level (1) CA-125 3. Quality of life (EHP-30) 4. Neuroangiogenic factor levels (1) bFGF (2) VEGF 5. Adenomyosis Evaluation Scale (1) Uterine volume 6. Pregnancy related indicators (1) Pregnancy rate (2) Miscarriage rate	1. (1) positive (*p* < 0.05) 2. (1) positive (*p* < 0.05) 3. positive (*p* < 0.05) 4. (1) positive (*p* < 0.05) (2) positive (*p* < 0.05) 5. (1) positive (*p* < 0.05) 6. (1) positive (*p* < 0.05) (2) positive (*p* < 0.05)	N.R.
Wu (2023) [29]	49/49	Shaofu Zhuyu Decoction + Dinoprostone	3 months	None	49/49	Dinoprostone	3 months	None	1. TER 2. Tumor maker levels (1) CA-125 3. Neuroangiogenic factor levels (1) bFGF (2) VEGF	1. positive (*p* < 0.05) 2. (1) positive (*p* < 0.05) 3. (1) positive (*p* < 0.05) (2) positive (*p* < 0.05)	N.R.
Zhang (2023) [30]	45/45	Shaofu Zhuyu Decoction + Progesterone	3 months	1 year	45/45	Progesterone	3 months	1 year	1. Pain index (1) VAS 2. Tumor maker levels (1) CA-125 3. Quality of life (EHP-30) 4. Neuroangiogenic factor levels (1) bFGF (2) VEGF 5. Adenomyosis Evaluation Scale (1) Uterine volume 6. Pregnancy related indicator (1) Pregnancy rate	1. (1) positive (*p* < 0.05) 2. (1) positive (*p* < 0.05) 3. positive (*p* < 0.05) 4. (1) positive (*p* < 0.05) (2) positive (*p* < 0.05) 5, (1) positive (*p* < 0.05) 6. (1) No significant difference	N.R.
Huang (2024) [31]	54/54	Modified Shaofu Zhuyu Decoction	3 months	None	54/54	Mifepristone	3 months	None	1. Pain index (1) VAS 2. TER 3. Tumor maker level (1) CA-125 4. Inflammatory factor levels (1) IL-6 (2) IL-8 (3) TNF-α 5. Immunological factor levels (1) Helper T Lymphocytes (2) Regulatory T Lymphocytes	1. (1) positive (*p* < 0.05) 2. positive (*p* < 0.0001) 3. (1) positive (*p* < 0.05) 4. (1) positive (*p* < 0.05) (2) positive (*p* < 0.05) (3) positive (*p* < 0.05) 5. (1) positive (*p* < 0.01) (2) positive (*p* < 0.01)	(1) SZD group: Mild diarrhea (1 case), Menorrhagia (1 case) (2) CM group: Palpitations (1 case), Mild dizziness (1 case)

AE: Adverse Event, bFGF: basic Fibroblast Growth Factor, CA-125: Cancer Antigen 125, CM: Conventional Medication, EHP: Endometriosis Health Profile, F/U: Follow-up, IL: Interleukin, N.R.: Not reported, PGE2: Prostaglandin E2, PGF: Prostaglandin, SZD: Shaofu Zhuyu Decoction, TER: Total Effective Rate, VAS: Visual Analog Scale, VEGF: Vascular Endothelial Growth Factor.

**Table 2 pharmaceuticals-18-01296-t002:** Characteristics of Shaofu Zhuyu Decoction in the included studies.

Study ID (Author, Year)	TCM Pattern Identification	Composition of Herbal Medicine	Modified Herbs	Dosage Details (Dosage, Frequency, Duration)	Duration
Zhang (2007) [21]	Blood Stasis *	*Angelicae Gigantis Radix* 12 g, *Cinnamomi Cortex* 3 g, *Cnidii Rhizoma* 10 g, *Corydalis Tuber* 10 g, *Foeniculi Fructus* 2 g, *Myrrha* 10 g, *Paeoniae Radix Rubra* 15 g, *Trogopterori Faeces (Fried)* 10 g, *Typhae Pollen* 10 g, *Zingiberis Rhizoma* 3 g	(1) For those with prolonged menstrual bleeding: Add *Artemisiae Argyi Folium*, *Fried Zingiberis Rhizoma*, *Leonuri Herba* (2) For Yang Deficiency *, Cold limbs, and Pulse that is deep and thin: Add *Cuscutae Semen*, *Processed Aconiti Lateralis*, *Morindae Officinalis Radix* (3) For Pelvic Masses: Add *Persicae Semen*, *Sparganii Rhizoma*, *Curcumae Rhizoma*, *Eupolyphaga*	1 dose per day, taken divided into two times (morning, dinner), start taking after discontinuing danazol, taken everyday	6 months + 3 months (if not treated)
Hong (2014) [22]	Blood Stasis *	*Angelicae Gigantis Radix* 12 g, *Cinnamomi Cortex* 5 g, *Cnidii Rhizoma* 10 g, *Corydalis Tuber* 15 g, *Foeniculi Fructus* 5 g, *Myrrha* 10 g, *Paeoniae Radix Rubra* 10 g, *Trogopterori Faeces (wrapped)* 10 g, *Typhae Pollen (wrapped)* 10 g, *Zingiberis Rhizoma* 5 g	(1) For Qi Deficiency *: Add *Codonopsis Radix* 10 g, *Atractylodes Macrocephalae Rhizoma* 9 g (2) For Pelvic Masses or Nodules: Add *Sparganium Rhizoma* 10 g, *Curcumae Rhizoma* 10 g, *Persicae Semen* 10 g, *Carthami Flos* 10 g, *Gleditsiae Fructus* 10 g (3) For those with prolonged menstrual bleeding: Add *Artemisiae Argyi Folium* 10 g, *Zingiberis Rhizoma (Fried)* 10 g, *Leonuri Herba* 10 g (4) For Deficient Essence and Blood: Add *Cuscutae Semen* 15 g, *Leonuri Fructus* 12 g, *Cistanches Herba* 12 g, *Piper wallichii* 10 g	1 dose per day, twice a day, taken every day (except menstruation period)	3 months
Liu (2020) [23]	Cold Coagulation Blood Stasis *	*Angelicae Gigantis Radix* 9 g, *Cinnamomi Cortex* 3 g, *Cnidii Rhizoma* 6 g, *Corydalis Tuber* 3 g, *Foeniculi Fructus* 7 g, *Myrrha* 6 g, *Paeoniae Radix Rubra* 6 g, *Trogopterori Faeces* 6 g, *Typhae Pollen* 9 g, *Zingiberis Rhizoma* 1 g	(1) Modifications Based on Menstrual Cycle ① Menstrual Phase: Add *Panax Notoginseng Radix*, *Rubiae Radix*, *Persicae Semen*, *Carthami Flos*, *Hirudo*, *Rhei Radix et Rhizoma (raw)*. Remove strong warming herbs. ② Follicular Phase: Add *Angelicae Gigantis Radix*, *Paeoniae Radix Alba*, *Cnidii Rhizoma*, *Rehmanniae Radix Preparata*, *Epimedii Herba*, *Morindae Officinalis Radix*. Remove strong blood-moving herbs. ③ Ovulatory Phase: Add *Bupleuri Radix*, *Cyperi Rhizoma*, *Corydalis Tuber*, *Leonuri Herba*, *Salviae Miltiorrhizae Radix*. Remove excessively warming herbs. ④ Luteal Phase: Add *Dipsaci Radix*, *Eucommiae Cortex*, *Cornu Cervi Degelatinatum*, *Epimedii Herba*, *Morindae Officinalis Radix*. Remove strong blood-moving herbs and cold herbs. (2) Modifications Based on Symptoms ① Patients with Pelvic Nodules and Masses: Add *Sargassum*, *Linderae Radix*, *Myrrha* ② In cases of heavy menstruation: Add *Notoginseng Radix*, *Rubiae Radix* ③ In cases of scanty menstruation: Add *Carthami Flos*, *Persicae Semen*, *Cnidii Rhizoma* ④ In cases with endometrial ectopic cysts: Add *Persicae Semen*, *Tabanus*, *Rhei Radix et Rhizoma (raw)*, *Hirudo* ⑤ In cases with abdominal bloating and pain: Add *Angelicae Dahuricae Radix*, *Toosendan Fructus* ⑥ In cases of nausea and vomiting: Add *Pinelliae Rhizoma*, *Citri Reticulatae Pericarpium* ⑦ In cases of cold-induced abdominal pain: Add *Citri Reticulatae Semen*, *Litchi Semen*	(1) 1~5 days of menstruation: 1 dose per day, a total of 400 mL taken divided into 3 times. (2) Otherwise: 1 dose per day, taken 2 times per day, 200 mL per time.	3 months
Zhang (2021) [24]	Cold Coagulation Blood Stasis *	*Angelicae Gigantis Radix* 12 g, *Cinnamomi Cortex* 5 g, *Cnidii Rhizoma* 6 g, *Corydalis Tuber* 15 g, *Foeniculi Fructus* 3 g, *Myrrha* 10 g, *Paeoniae Radix Rubra* 10 g, *Trogopterori Faeces* 10 g, *Typhae Pollen (wrapped)* 10 g, *Zingiberis Rhizoma* 3 g	(1) For Severe Abdominal Pain: Add *Achyranthes Bidentata* 10 g, *Veratrum Formosanum* 10 g, *Draconis Sanguis* 3 g (take separately) (2) For Pelvic Nodules or Masses: Add *Persicae Semen* 9 g, *Curcumae Rhizoma* 9 g, *Sparganium Stoloniferum* 10 g, *Gleditsiae Spina* 10 g (3) For Excessive Menstruation and Blood Clots: Add *Agrimoniae Herba* 20 g, *Charred Rubiae Radix* 15 g, *Sepiae Endoconcha* 15 g, *Panax Notoginseng Radix* 3 g (take separately)	1 dose per day, twice a day, taken every day (except menstruation period)	3 months
Chen (2023) [25]	Not reported	*Angelicae Gigantis Radix* 9 g, *Cinnamomi Cortex* 12 g, *Cnidii Rhizoma* 10 g, *Corydalis Rhizoma* 12 g, *Foeniculi Fructus* 12 g, *Myrrha* 10 g, *Paeoniae Radix Rubra* 15 g, *Trogopterori Faeces* 10 g, *Typhae Pollen (Fried)* 15 g, *Zingiberis Rhizoma* 9 g	None	200 mL per time, twice a day, taken from the 3rd to the 5th day after menstruation begins.	3 months
Dai (2023) [26]	Not reported	*Angelicae Gigantis Radix* 12 g, *Cinnamomi Cortex* 5 g, *Cnidii Rhizoma* 6 g, *Corydalis Tuber* 15 g, *Foeniculi Fructus* 3 g, *Myrrha* 10 g, *Paeoniae Radix Rubra* 10 g, *Trogopterori Faeces* 10 g, *Typhae Pollen (wrapped)* 10 g, *Zingiberis Rhizoma* 3 g	(1) For Abdominal Pain: Add *Achyranthes bidentata* 10 g, *Veratrum formosanum* 10 g, *Draconis Sanguis* (take separately) 3 g (2) For Pelvic Nodules, Masses: Add *Sparganium stoloniferum* 10 g, *Gleditsiae Spina* 10 g, *Prunus persica* 9 g, *Curcuma zedoaria* 9 g (3) For Excessive Menstruation: Add *Agrimoniae Herba* 20 g, *Charred Rubiae Radix* 15 g, *Sepiae Endoconcha* 15 g, *Panax notoginseng Radix* (take separately) 3 g	1 dose per day (total 300 mL per day), 150 mL per time, twice a day	3 months
Hu (2023) [27]	Cold Coagulation Blood Stasis *	*Angelicae Gigantis Radix* 9 g, *Cinnamomi Cortex* 12 g, *Cnidii Rhizoma* 10 g, *Corydalis Tuber* 12 g, *Foeniculi Fructus* 12 g, *Myrrha* 10 g, *Paeoniae Radix Rubra* 15 g, *Trogopterori Faeces* 10 g, *Typhae Pollen* 15 g, *Zingiberis Rhizoma* 9 g	None	150 g per time, twice a day, taken during menstruation	3 months
Wen (2023) [28]	Not reported	*Angelicae Gigantis Radix* 15 g, *Cinnamomi Cortex* 3 g, *Cnidii Rhizoma* 10 g, *Corydalis Tuber* 10 g, *Foeniculi Fructus* 3 g, *Paeoniae Radix Rubra* 15 g, *Trogopterori Faeces* 10 g, *Typhae Pollen* 10 g, *Zingiberis Rhizoma* 3 g	(1) For Severe Cold Dampness: Remove *Zingiberis Rhizoma*, Add *Zingiberis Rhizoma Praeparatum*, *Evodiae Fructus* (2) For Severe Spleen and Kidney Yang Deficiency: Add *Eucommiae Cortex*, *Codonopsis Radix* (3) For Liver Qi Stagnation *: Add *Curcumae Zedoariae Rhizoma*, *Bupleuri Radix* (4) For Excessive Menstruation and Blood Clots: Add *Panax Notoginseng*, *Rubiae Radix* (5) For Severe Blood Stasis *: Add *Spargani Rhizoma*, *Curcumae Rhizoma* (6) For Spleen Deficiency: Add *Atractylodis Macrocephalae Rhizoma Praeparatum*, *Atractylodis Rhizoma*	1 dose per day, a total of 300 mL taken divided into two times (morning, dinner), taken every day	3 months
Wu (2023) [29]	Blood Stasis *	*Angelicae Gigantis Radix* 15 g, *Cinnamomi Cortex* 6 g, *Cnidii Rhizoma* 12 g, *Corydalis Tuber* 6 g, *Foeniculi Fructus (Fried)* 3 g, *Myrrha* 12 g, *Trogopterori Faeces (Fried)* 12 g, *Typhae Pollen* 15 g, *Zingiberis Rhizoma* 6 g	(1) For patients with severe anemia: Add *Astragali Radix* 20 g, *Angelicae Gigantis Radix* 5 g. (2) For patients with excessive menstrual bleeding and prolonged bleeding: Add *Charred Cirsii Herba* 6 g, *XCrinis Carbonisatus* 6 g. (3) For patients with Qi deficiency: Add *Polygonati Rhizoma* 15 g, *Astragali Radix* 15 g.	1 dose per day, a total of 400 mL taken divided into two times (morning, dinner)	3 months
Zhang (2023) [30]	Cold Coagulation Blood Stasis *	*Angelicae Gigantis Radix* 15 g, *Cinnamomi Cortex* 3 g, *Cnidii Rhizoma* 10 g, *Corydalis Tuber* 10 g, *Foeniculi Fructus* 3 g, *Paeoniae Radix Rubra* 15 g, *Trogopterori Faeces (wrapped)* 10 g, *Typhae Pollen (wrapped)* 10 g, *Zingiberis Rhizoma* 3 g	(1) For Severe Spleen and Kidney Yang Deficiency *: Add *Codonopsis Radix*, *Eucommia Cortex* (2) For Severe Cold and Dampness: Remove Zingiberis Rhizoma, Add *Zingiberis Rhizoma Praeparatum*, *Evodiae Fructus* (3) For Severe Blood Stasis *: Add *Sparganium Rhizoma*, *Curcumae Rhizoma* (4) For Excessive Menstruation and Blood Clots: Add *Rubiae Radix*, *Panax Notoginseng Radix* (5) For Liver Qi Stagnation *: Add *Bupleuri Radix*, *Curcumae Radix* (6) For Spleen Deficiency: Add Fried *Atractylodes Rhizoma*, *Atractylodes Macrocephalae Rhizoma (Fried)*	150 mL per time, twice a day, every day (except menstruation period)	3 months
Huang (2024) [31]	Shen Deficiency Blood Stasis *	*Angelicae Gigantis Radix* 18 g, *Cinnamomi Cortex* 6 g, *Cnidii Rhizoma* 10 g, *Corydalis Tuber* 15 g, *Foeniculi Fructus* 6 g, *Myrrha* 10 g, *Paeoniae Radix Rubra* 12 g, *Trogopterori Faeces* 9 g, *Typhae Pollen* 15 g, *Zingiberis Rhizoma* 6 g, *Curcumae Rhizoma* 15 g, *Eupolyphaga* 10 g, *Fluoritum* 20 g, *Sparganii Rhizoma* 9 g	None	1 dose per time, twice a day, a total of 10 doses, from 5 days before menstruation to the start of menstruation.	3 months

* “Qi stagnation” describes a pathological state in which the flow of qi is impaired or blocked, leading to symptoms such as pain, distension, irritability, and mood disturbances; “Qi deficiency” refers to a state where there is an insufficiency or weakness of qi, resulting in fatigue, shortness of breath, spontaneous sweating, weak voice, and general lack of energy; “Yang deficiency” describes a pattern characterized by insufficient Yang qi (functional energy), leading to hypofunction of organs, impaired warmth, and decline in metabolic activities; “Blood stasis” describes a pathological condition caused by impaired or obstructed blood circulation, leading to blood accumulating in a specific area.

**Table 3 pharmaceuticals-18-01296-t003:** Characteristics of control group treatments in the included studies.

Study ID (Author, Year)	Treatment	Dosage Details (Dosage, Frequency)	Special Notes	Duration
Zhang (2007) [21]	Danazol	200 mg, 1–3 times daily	Take 3 times a day for the first 2 months, 2 times a day for the next 2 months, and once a day for the following 2 months.	6 months
Hong (2014) [22]	Progesterone	2.5 mg, twice weekly	None	3 months
Liu (2020) [23]	Gestrinone	5 mg, once daily	None	3 months
Zhang (2021) [24]	Mifepristone	25 mg, once daily (except menstruation period)	None	3 months
Chen (2023) [25]	Ibuprofen	400 mg, twice daily, taken from the 3rd to the 5th day after menstruation begins.	None	3 months
Dai (2023) [26]	GnRH-a or Mirena ring	(1) In cases of surgical treatment: GnRH-a Triptorelin Acetate 75 mg, once every 28 days (2) In cases without surgical treatment: Insert the Mirena ring into the uterine cavity	None	3 months
Hu (2023) [27]	Ibuprofen	200 mg, up to four times daily	Take every 4 to 6 h if pain or fever persists during the menstrual period.	3 months
Wen (2023) [28]	Progesterone	2.5 mg, twice weekly		3 months
Wu (2023) [29]	Dinoprostone	2 mg, once daily (except menstruation period)	Take from the 6th day of menstrual cycle.	3 months
Zhang (2023) [30]	Progesterone	2.5 mg, twice weekly	None	3 months
Huang (2024) [31]	Mifepristone	25 mg, once daily (except menstruation period)	None	3 months

## Data Availability

The raw data supporting the conclusions of this article will be made available by the authors on request.

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
