# Peer review of "Shaofu Zhuyu Decoction for Treating Endometriosis: A Systematic Review and Meta-Analysis"

_pharmaceuticals, 2025, doi:10.3390/ph18091296_

Round 1
Reviewer 1 Report
Comments and Suggestions for Authors
improve introduction :
- The transition between sections (from modern treatment to TCM, and then to SZD studies) could be smoother. Consider using clear sub-themes or transitional sentences.
-For example: “Given the limitations of current treatments, there is growing interest in complementary therapies, particularly Traditional Chinese Medicine (TCM).
-Some statements feel repetitive or overly detailed for an introduction.
E.g., listing every theory of pathogenesis might overwhelm the reader; summarizing them briefly and citing could be more concise.
Typo: "ndometriosis" (missing "E" at the beginning) should be fixed.
- Tble : i qdvice to change orientation of table
Author Response
We sincerely appreciate the reviewer’s time and effort in evaluating our manuscript. We have addressed the comments point by point and revised the manuscript accordingly. The revised version is attached for your kind consideration.

Reviewer 2 Report
Comments and Suggestions for Authors
The manuscript by Su-Bin Kim and co-authors presents a systematic review and meta-analysis examining the efficacy and safety of Shaofu Zhuyu Decoction (SZD) for endometriosis treatment. The topic is clinically relevant in view of the limitations of conventional endometriosis treatments and growing interest in complementary therapies. The review follows appropriate methodology with PRISMA guidelines and PROSPERO registration, which enhances its credibility. The risk of bias assessment using the Cochrane RoB 2.0 tool is appropriately conducted and correctly reported. The authors have conducted a valuable review of an important topic, and the performed analysis will be interesting to the journal's readership.
The most significant concern is the pooling of studies with heterogeneous control interventions without adequate subgroup analysis. The control groups used seven different conventional medications (progesterone, mifepristone, ibuprofen, danazol, dinoprostone, gestrinone, and GnRH-a/Mirena), which have different mechanisms of action and efficacy profiles. Combining these in a single meta-analysis without subgroup analysis by control medication type substantially limits the interpretability of the results.
Page 18: The conclusion that SZD "may not only alleviate clinical symptoms but also target the underlying disease mechanisms" is somewhat speculative given the nature of the evidence.
Summarizing, I recommend major revision of the manuscript before acceptance.
Author Response

(The authors gave the same response as above.)

Reviewer 3 Report
Comments and Suggestions for Authors
In the review by Park et al. entitled »Shaofu Zhuyu decoction for Treating Endometriosis: A Systematic Review and Meta-Analysis«, the authors provide an overview of scientific studies dealing with the traditional Chinese herbal formula Shaofu Zhuyu Decoction (SZD), which is frequently used for gynecological diseases with blood stasis and abdominal pain. The systematic review and meta-analysis evaluates the efficacy and safety of SZD in combination with conventional medication (CM) in the treatment of endometriosis. Eleven randomised controlled trials (n = 1,186) were analysed. The combination of SZD and CM significantly improved the overall efficacy rate (OR = 1.15), reduced serum CA-125 levels (OR = –1.57), and relieved pain (OR = –4.90) compared to CM alone. The recurrence rate was lower and the side effects were mild and comparable. Despite the promising results, there are limitations such as methodological heterogeneity and geographical concentration of the studies. Further multicenter trials are needed to validate the integration of SZD into global endometriosis treatment.
The English language in the article is generally easy to read and understand, especially for readers who are familiar with medical terminology. However, there are several areas where the quality of the language could be somewhat improved to increase clarity and flow – for example, by shortening excessively long sentences or making transitions from one section to the next more fluid. Nevertheless, the paper is still readable in its current form.
Although the paper, as a review, does not contain any original findings obtained through direct research activity (new experimental data), it does have some elements of originality. The most important element is that the study is the first systematic review of SZD that comprehensively evaluates its clinical efficacy and safety in the treatment of endometriosis. As the authors themselves acknowledge, several meta-analyzes of SZD have already been published, but these have primarily focused on examining the combination therapy of SZD and conventional medicine. However, it is difficult to draw firm conclusions from such studies about the isolated effect of SZD monotherapy. One of the original contributions is also the fact that the study also reports on clinical outcomes, which adds to the quality of the review. The weak point of the study could be that most of the cited papers were written by scientists of Chinese origin. This is hardly surprising considering that Shaofu Zhuyu Decoction is a traditional Chinese herbal formula that is virtually unknown outside of Chinese culture. Thus, all 11 randomized controlled trials that were eventually included as references were conducted in China and published in Chinese, which is of course a limitation of this study, but is due to the nature of the study.
Considering the suitability of the paper for publication in Pharmaceuticals, I must say that the PRISMA guidelines were followed in the selection of suitable original research, which in principle should guarantee an appropriate selection of literature on which to base the review of the paper. Indeed, it seems to me that the authors have made a notable effort in the selection of suitable original papers. Furthermore, the prevalence of endometriosis is quite high, and affects the female population worldwide, which is another factor in favor of publishing the paper. Although the use of SZD is not expected to solve the problem of endometriosis, the results of the study suggest that the use of SZD may have a therapeutic effect in the treatment of endometriosis through an integrative approach alongside conventional hormonal or pharmacologic medications. Although not a groundbreaking study, the quality of the work is high enough to be published in Pharmaceuticals.
In view of all that has been written above, I recommend publishing the article after a minor revision, taking into account the following two comments:
1.) In the paper, the terms “qi stagnation” and “qi deficiency” are mentioned but not explained. I believe that people who are familiar with Chinese culture may know these terms, but there are also people who are not acquainted with them. Therefore, please explain these terms.
2.) Correct the grammatical error “decotion” (it should be “decoction”) in the manuscript.
Author Response

(The authors gave the same response as above.)

Round 2
Reviewer 2 Report
Comments and Suggestions for Authors
The revised version of the manuscript was significantly improved by the authors. The comments were addressed properly. I recommend acceptance of the manuscript for publication in the revised form.